# Regional Patterns and Hierarchical Tendencies: Analysis of the Network Connectivity of 63 Service-Oriented Tourist Cities in China

**Hongjiu Tang**

School of Geography and Planning, Geography and Environment Building, Sun Yat-Sen University, Guangzhou 510275, China; tanghj5@mail2.sysu.edu.cn

**Abstract:** Previous studies of service-oriented tourist city networks have often focused on the analysis of the geographical distributions and service roles of important cities instead of the connections and hierarchical tendencies between different types of cities within a whole region. The current study uses big data approaches for the regional connections of 38 tourism organizations, including famous hotels, air passenger transport services, and tourism service agencies, across 63 of the most important tourist cities in China. Fuzzy c-means clustering analysis is used to define eight city arena clusters. According to the distributions of connectivity between the 63 cities, these eight clusters play different functional service roles in the urban tourism network in four hierarchies. With their "center–edge" memberships, these arena clusters are formed by the interweaved process of regional and hierarchical tourism service connections. The results here include analysis of the various service-oriented tourist cities in China and point out the geographical "gap" faced by networks. Service-oriented tourist cities need to find their hierarchies and positioning in the network, scientifically speaking, to avoid blind development and to support sustainable regional tourism development in urban areas.

**Keywords:** tourism geography; city connectivity; tourism service value; functional role; urban hierarchy; regional pattern

---

## 1. Introduction

One of most important academic viewpoints in Manuel Castells' theory points out that "the deep combination between the information technology revolution and capitalist reorganization constructs the flow network social space. The city is a spatial unit of labor reproduction or a space fragment in the network society, and the network society is composed of multiple cities" [1]. He also thinks that network society relates to such practice phenomena where important tourist cities across some specific large regions are used by different capitals as "basing points" in the connection networks of service and production [2]. The resulting connections make it necessary to promote the development of important cities and arrange some service-oriented tourist cities into a connected network hierarchy. However, the lack of theoretical agreement on the defining characteristics of tourist cities in Manuel Castells' theory has resulted in scientific taxonomies, which are usually limited to focusing on the highest cities in the hierarchies [3]. Except for the lack of an undisputed definition of a service-oriented tourist city itself, the main reason for these eclectic approaches stems from the lack of systematic and reliable data [4], which is, of course, related to the absence of undisputed definition characteristics of service-oriented tourist cities. One of the main consequences of this problem is associated with Friedman's description of complex spatial hierarchies, i.e., cities that are lower in rank or appear less important are still not evaluated in this transnational or regional hierarchy [5].

However, many studies that focus on tourist cities have not considered the service functions and hierarchical differences of various service-oriented tourist cities in urban tourism networks from a large

whole-region perspective, which results in the lack of scientific positioning and reasonable development strategies for service-oriented tourist cities of different hierarchies. In fact, different tourist cities will have varying degrees of frequent or minimal connection in the urban tourism network. The reason for these differences is that various cities have different functional nodes (such as "basing points") in the network, and the corresponding cities have different service function types and service scopes. This difference essentially comes from the competition with the abilities to gain capital among other cities, and this characteristic can be reflected by the hierarchical tendency in the network. For example, Beijing, as an international service-oriented tourist city, is the capital of China. It is the urban functional center of China's political management, cultural production, high educational supply, and passenger transport hub. These important urban functions most likely cause many tourists to prefer to travel to Beijing. Compared with Beijing, although Guilin has high-quality natural landscape tourism resources, it does not have many types of urban functions, and the city's financial function, exhibition function, cultural production, passenger transportation, and other service functions are not powerful. Under the restriction of the city's service functions, it cannot attract many tourists with high-end tourism preferences to flow into the city. This shows that only relying on high-quality tourism resources cannot drive Guilin to obtain considerable economic benefits from tourism, and this kind of city (where tourism service functions are not powerful) does not occupy an important hierarchical position in the tourism connection network.

Various scholars have empirically attempted to use the interlocking network model (INM) designed by Taylor [6] to analyze service-oriented tourist city networks. This model has been widely studied and applied in service-oriented tourist city connection networks from the perspective of regionalization. Because the connection network is closely related to the geographical distribution of cities, the novelty of the study must be in attempting to describe and explain the connected model of an interlocking network based on tourism services from the geographical perspective, and in pointing out the geographical "gap" faced by service-oriented tourist city networks. In terms of methodology, the research team found that there are some problems in the previous cluster analysis. For example, due to the dynamic development of tourism, some cities do not fixedly belong to a fixed cluster. Therefore, one of the specific novelties is the use of fuzzy c-means clustering analysis to solve these problems. We find that the interlocking network model of tourism services also shows the "center–edge" geographical structure in reality, but it is a new "center–edge" mode. The other specific novelties lie in the following aspects: First, the new "center–edge" model is a polycentric three-dimensional connectivity network; central cities and edge cities have different hierarchical tendencies, and they also have different functional roles in the network. Second, the new "center–edge" model has a bidirectional flow; unlike cities in the traditional model, central and edge cities can provide tourism services, products, and tourists with each other. Therefore, the new model in this paper would enrich the application of "center–edge" theory in new research scenarios. The research question of this paper is determining how to scientifically divide the hierarchical tendencies and roles of tourist cities in the tourism service connection network from a regional perspective, and, thus, to point out the regional patterns between cities of different clusters and their hierarchies for future regional tourism policy formulation.

Through the efforts of the researcher, this paper analyzes the network connectivity characteristics of tourist cities with different hierarchical tendencies from the perspective of the overall geographical region so as to improve the division and evaluation field of city connectivity in tourism geography research. Here, the main research objectives of this paper are as follows: On the one hand, summarizing the fuzzy spatial dimensions behind the formation of tourist cities, and, on the other hand, trying to describe the geographical details of the network in order to illustrate that geographical space is an important factor in the formation of the network. In the realistic process of tourism management, understanding the flow network structure of tourist cities with different hierarchical tendencies in a region can help to formulate the tourism development strategies for different ranks of cities and to promote the healthy development of regional tourism.

## 2. Literature Review

### 2.1. The Hierarchy of Tourist Cities and the Flow Space Theory of Castells

The term "service-oriented tourist city" refers to a city with natural or artificial tourism attractions and a relatively complete tourism service reception capacity, where the tourism service industry accounts for a certain proportion of the city's income and can provide corresponding services for foreign tourists with different demand levels [7]. Therefore, an urban tourism network is defined as an organic aggregate of various tourism service organizations pursuing regional location strategies. In this way, relying on the representative tourism service organizations, small- and medium-sized tourist cities in different geographical locations as well as large tourist cities together form an interlocking network structure. Through mutual tourism connections, each service-oriented tourist city can become a tourism service provider in the network [8]. A network of service element flows is an integrated network formed by various elements, such as service capital flow, tourist flow, information flow, communication, and interaction with tourism companies [2]. In short, service element flow is the leading activity of the network in tourist city society, which will lead to different service levels in urban tourism areas with various connections, and this also reflects the differences between tourism service functions and the sizes of tourist cities in a specific region. Castells also pointed out that the characteristic of the flow space is "a material organization that operates with the characteristic of flow and relies on various pipelines or technical networks to share the synchronicity of social practice" [1]. This kind of convergence traditionally relies on the physical proximity, that is, the simultaneity of the presence, which needs to connect simultaneously between various cities in the network society. According to Castells' theory, the network structure constructed by tourism service connections of tourist cities mainly includes three aspects: First, the connection and communication between tourism service industries; second, different tourism service functions, such as guiding nodes, service bases, or tourist centers from different cities in the network; and third, the spatial structure of the dominant central cities and some edge cities [9].

### 2.2. The Interlocking Network Model (INM)

The interlocking network model (INM) originated from early scholars' criticisms of the conceptual and empirical defects in the city network literatures. The interlocking network model has three levels: The network level, including various cities at different levels, which are connected in their regional tourism economy; the node level, where some cities play a pivotal role in a specific region; and the sub-node level, including senior tourism organizations providing different specific services for tourists [10]. Beaverstock (2000) argued that many of these concepts only focus on measuring the attributes of tourist cities, ignoring the importance of connected relationships within regional urban systems [11]. David and Michael (2001) then improved the INM by specifying the "service-oriented tourist city network" based on the relationship between cross-regional leading organizations and other internal tourism service organizations [12]. In a word, the INM provides scholars with a specific solution to the problem of network relationships. Although it is difficult to obtain comprehensive data for the flow relationships between cities in practical research, the urban interlocking network structure can be measured by the connected relationships between cities.

Today, the INM uses samples from different tourism organizations to apply urban tourism research at the regional, national, and global scales. Some scholars use the subsidiaries of famous international companies in different cities to judge the frequency of tourism communication between the subsidiaries and the parent company [13]. Then, research teams can further determine the functional value of the city in the tourism connection network. These types of companies include airlines, large hotels, financial services offices, e-tourism service platforms, etc. [14].

### 2.3. The Connections between Various Service-Oriented Tourist Cities and Their Hierarchical Tendencies

The contemporary study of tourist cities started with professor Bao Jigang [15,16] with the identification of a "center city" to manage and control the "regional tourism service market segment" created by multiregional tourism organizations. This theory reflects Wise's recognition of the transition from an individual urban economy to a regional economy, which is characterized by the inevitable development trend of an increasingly integrated regional tourism network [17]. Tourist cities are the basing points for a connected network, so their specifications are related to the identification of a "city in the tourism relationship matrix within the region", as the *Travel and Tourism Competitiveness Report 2017* [18] shared with us. Therefore, the relevant discussion focuses on the "regional urban tourism network" [19], a "transnational urban tourism system" [20], "functional service-oriented tourist city system", or a "global urban tourism network" [21]. The integration of these different concepts has never been sufficiently analyzed and discussed regarding this aspect in the literature for service-oriented tourist cities.

The rise of the tourism connection network is one of the most fundamental changes in the current tourism development according to the *Travel and Tourism Competitiveness Report 2019* [22], where, for a long time, city economic bases have been transforming from the industrial manufacturing industry to high-value-added service industry. This self-accelerative transformation is reflected in the emergence of more and more types of tourism services in important world cities [13], which are unable to cope with the increasing pressure of regional urban structural changes and innovation of tourism service products. More and more cities may gradually lose their attraction, and their tourism influence within the regional tourism network may decline. The important point here is that these tourism services are an indispensable factor of city services, which have their own growth potential. Compared with other fields of urban tourism service sector growth, this rapid growth is also the product of the interacting results of demands derived from other sectors [23]. The reason for this is that tourism service organizations in these cities will benefit immensely from advances in communication, information, and virtualization technology, which will enable them to broaden the spatial scope of tourism services. For example, tourism organizations are generally related to the characteristics of tourism demand groups in specific cities, such as an "air tourism organization" or "tourism hotel management alliance", etc.; however, under the conditions of contemporary globalization, due to the huge tourist scale advantage in China, tourism organizations can easily realize "small profits and quick turnover". Some cities' tourism organizations choose to implement various chain service alliance strategies in China, and there are many tourism subsidiaries that rely on standardized services to gain a competitive advantage [24]. Based on these observations, this paper focuses on the tourism service process between various cities, and enables testing of the proposition that there is a complex tourism service that has distinct characteristics pertaining to the geographical location and regional patterns.

These kinds of tourism services require the cooperation and convenience brought about by the distribution of various tourism service organizations. Therefore, the headquarters of tourism service organizations may bring more "tourism service value" than the subsidiaries. However, the headquarters of tourism service organizations usually require the service-oriented tourist city to have a better location in the urban tourism network [25]. In this context, different headquarters of tourism service organizations tend to aggregate in large cities with perfect urban functions that have a wide range of influences. These cities often have the highest hierarchical tendencies in the tourism service network, and they have a leading role with respect to their surrounding cities [14]. The higher the hierarchical position of a service-oriented tourist city, the more effective it is at neutralizing the negative influence of distance barriers when tourists make tourist destination decisions [13]. For the consideration of location strategy and tourism externality of tourist cities with high positions in the hierarchy, there will be different ranks of cities around the leading service-oriented tourist city in a region. They have different service divisions and urban functions around the tourism service industry chain, and their spatial distribution structure within the region will form a tourism connection cluster [26].

### 2.4. Advanced Tourism Service Organizations and the Representation of City Connection Hierarchy

The most important tourism fields that can reflect the value of a city's tourism services are the following: The distribution of famous hotels at home and abroad [27]; the service capacity of air passenger transport for tourism [28]; and the distribution of tourism consulting service agencies [29]. These representative tourism organizations can effectively compare the tourism service value with various tourist cities, and these three types of organizations can also reflect the real competitive differences between different tourist cities [30]. Drawing on the research achievements of scholars from various fields regarding the role of advanced tourism services in the formation of a city network, the Globalization and World Cities Research Network (GaWC) has theoretically explained the collection of connected data (http://www.lboro.ac.uk/gawc). According to the GaWC's collection principles, this paper treats service-oriented tourist cities as tourism service centers in specific regional networks, and a method is developed here to analyze and measure the networks of service-oriented tourist cities [31]. The method chooses the affiliation type (head office or subsidiary) of various advanced tourism organizations in different tourist cities [23]. Then, there must be some tourism connection between the tourism head office and the tourism subsidiary, such that the city can be given a certain score; for example, a city with a head office is recorded as having five points, while a city with a general subsidiary is recorded as having two points. Finally, according to the city scores of different advanced tourism organizations in some given cities, the scores of several advanced tourism organizations in a city can be obtained in order to reflect the total score of the service-oriented tourist city in the whole connection network, and the total scores of these cities can reflect differences in the importance of different cities regarding urban tourism connections [32]. This method does not assume that service-oriented tourist cities will form an obvious city hierarchy, but instead designates a tourism network in a region according to the connections between different advanced tourism industries in various cities, where the "hierarchical tendencies" can then be revealed [6]. Based on previous urban tourism network research, this can bring two advantages: First, the advanced tourism industry is widely represented. The data are relatively easily obtained due to various connections and exchanges between various cities. The data size needed is quite huge, and this can solve the problem of research data deficiency to a great extent. Secondly, the urban tourism network brings a very large number of service-oriented tourist cities into the connection network for analysis. The analysis of the relationship between multiple cities in the whole region is closer to reality; therefore, an overall understanding of the city connections in the tourism region can be acquired. However, before these results are described in more detail, this empirical principle requires the conceptual problems to be detailed.

### 2.5. Conceptualization: Service-Oriented Tourist Cities as Regional Centers

The concept of an important service-oriented tourist city as a regional center has been developed in the GaWC's research [6]. A service-oriented tourist city can be regarded as the center of a regional tourism service network, and various types of tourism service organizations (headquarters or subsidiaries) in the network focus on providing services for tourists with different travel purposes and consumption levels. Consequently, the tourism service connection network can be formally specified as an interlocking network model (INM). The formation of a service-oriented tourist city network is carried out at three levels through their cooperation with different levels of the network, where seamless connection services can be provided for tourists across different geographical locations of the whole region.

The connectivity of an urban tourism network can be formally expressed by the matrix Vij, defined by m cities × n tourism organizations, where $V_{ij}$ is the "tourism service value" of city i to tourism organization j [9]. The value of tourism services reflects the importance of a city to the relevant tourism organizations of the tourism service network. Therefore, every column denotes a tourism organization's regional layout strategy, and every row describes each city's mix of tourism services. These allow for two types of research. The focus column will describe tourism organizations and the focus row will provide knowledge regarding various tourist cities.

## 3. Methodology

### 3.1. Case Selection

China currently has 393 prefecture-level cities (as of December 31, 2019), but there are not many service-oriented tourist cities. The analysis of cities should not be carried out with as many cities as possible, because as the size of the relational data matrix increases (i.e., as more cities are added), the data may become relatively "sparse" (many zero entries), which reduces the reliability of the analysis. Thus, the criterion for city selection here was a network connectivity of at least one-twelfth the network connectivity of the highest city. In the initial work, 70 cities were assessed as "famous service-oriented tourist cities", combined with some tourist cities at the city scale, urban tourism representativeness, and other aspects of the existing problems. For example, Yangshuo is a county-level service-oriented tourist city that is well-known at home and abroad, but this paper selected cities above the prefecture level that have a larger administrative scale. Dunhuang is also a world-famous tourist city, but it is not a service-oriented tourist city, mainly based on the history and culture, and the characteristics of the urban tourism service industry are not obvious. After a strict selection process, a "roster" of 63 service-oriented tourist cities was devised. Here, 60 service-oriented tourist cities were selected from the statistics department of the Ministry of Culture and Tourism of the People's Republic of China (the most authoritative tourism management department in China), representing the important regional compositions of the urban tourism industry in mainland China. At the same time, this paper has also added three cities, including Hong Kong, Macao, and Taipei, because there are also obvious tourism service connections between these cities and the other 60 cities. These 63 cities have a good tourism reputation and a complete tourism service industry, and the income of urban tourism occupies an important part of the cities' local economic income. Therefore, this paper includes a total of 63 samples (Table 1).

**Table 1.** List of the 63 key tourist cities in China.

| Geographical Region | City Name | Number |
|---|---|---|
| North China | 1. Beijing, 2. Tianjin, 3. Shijiazhuang, 4. Hohhot, 5. Qinhuangdao, 6. Chengde, 7. Datong, 8. Taiyuan, 9. Luoyang; | 9 |
| East China | 10. Shanghai, 11. Nanjing, 12. Hangzhou, 13. Qingdao, 14. Ningbo, 15. Wenzhou, 16. Xiamen, 17. Suzhou, 18. Wuxi, 19. Nantong, 20. Lianyungang, 21. Jinan, 22. Yantai, 23. Weihai, 24. Hefei, 25. Fuzhou, 26. Huangshan, 27. Quanzhou, 28. Zhangzhou; | 19 |
| South China | 29. Guangzhou, 30. Shenzhen, 31. Zhuhai, 32. Shantou, 33. Zhanjiang, 34. Zhongshan, 35. Nanning, 36. Haikou, 37. Sanya, 38. Guilin, 39. Beihai; | 11 |
| Central China | 40. Wuhan, 41. Changsha, 42. Nanchang, 43. Zhengzhou, 44. Jiujiang, 45. Zhangjiajie; | 6 |
| Northeast China | 46. Shenyang, 47. Dalian, 48. Changchun, 49. Harbin, 50. Jilin; | 5 |
| Northwest China | 51. Xi'an, 52. Lanzhou, 53. Xining, 54. Yinchuan, 55. Urumqi; | 5 |
| Southwest China | 56. Chongqing, 57. Chengdu, 58. Kunming, 59. Guiyang, 60. Lasa; | 5 |
| Hong Kong, Macao, and Taiwan regions of China | 61. Taipei, 62. Hong Kong, 63. Macao. | 3 |

Note: In addition to the three tourist cities in Hong Kong, Macao, and Taiwan, the other 60 cities belong to the key tourist cities monitored by the Chinese tourism administration.

### 3.2. Data Production

To make these data more adherent with the general judgment of the expert team on the hierarchical tendencies of China's service-oriented tourist cities, the important problem is the inherent subjective response in the process of data creation, where the obtained data do not have the key attribute of intersubjectivity. In other words, the use of the same data does not always determine the same boundary results. Because of this, a key problem needs to be faced, that is, due to the large uncertainty of the created data, will there be irreparable credibility defects in the empirical analysis? There are two ways to reduce the impact of this problem: First of all, the scoring method should be designed to be as

simple as possible, with the principle of "0 means that the city does not have a specific organization (one of the 38 typical tourism organizations), 2 means that the city has a subsidiary company of a specific organization, and 5 means that the head company of a specific organization is in the city" to count the distribution types of each tourism service organization in each city [9]. Therefore, the subjective decision-making evaluation of organization types in this paper is limited to some relatively simple boundaries. Secondly, the empirical analyses should be carried out with a very large number of tourism organizations (as many as 38 tourism organizations), such that it is possible to iron out the accidental differences of specific individuals in the overall analysis designed for the data. Finally, like all data production methods, the generated relational data value distribution should have good credibility.

With the production of data here, some tourism organizations have very detailed information, while others have much less information. Therefore, by designing a relatively simple scoring system to accommodate the collection of multifarious information and by selecting measurement items with the same statistical caliber, the tension of an unequal data distribution can be solved. Using a six-point scale (0–5), two levels can be given easily, where a score of zero denotes no specific tourism organization in the city, and cities housing the headquarters of tourism organizations have a score of 5. Therefore, the key point of the scoring decision is allocating the middle four scores (1, 2, 3, and 4) in order to describe the tourism service values of various cities. This means that three boundaries must be specified for each tourism organization between points 1 and 2, 2 and 3, and 3 and 4 [33]. Therefore, the basic strategy of this paper was to use a score of 2 for cities assumed to have non-tourism headquarters (i.e., sub-organizations of headquarters). This score represents the "typical" or "normal" service level of a given tourism organization in a city. To determine this normality, one must find the overall average of the organization's distribution across all tourist cities. However, sometimes, some travel service organizations do not reach a given "normal" or "typical" level. For example, if a tourism organization's service is shared by other cities and its service scope for a single city is actually small, the corresponding tourism organization in that city will be scored as 1. A tourism organization in a city showing very few (or, perhaps, no) professional tourism services or tourism participants would also be scored as 1. Generally speaking, the boundary between 2 and 3 is based on the size factor of the tourism organization service scale, while the boundary between 3 and 4 depends on the extraterritorial factors of the organization layout type. For example, a super-large tourism organization type with many employees will lead to a score of 3 in the city where the organization is located, while a tourism organization with a regional headquarters will lead to a city score of 4. In fact, if possible, in order to determine the boundary scientifically, it is necessary to consider a mixture of the service scale and extraterritorial information when deciding the boundaries for each tourism organization in various cities. The end result is the service value matrix V, which is an m × n matrix data group. Therefore, the matrix data group in this paper has a 63 × 38 specific data array, with $V_{ij}$ ranging from 0 to 5.

### 3.3. Data Collection

Precise specifications guided the data collection process here; the research team completed the task of data collection after six days of work. The data collection method is described in detail by the collection method of Taylor et al. [33]. In this paper, a representative advanced service tourism organization was defined as a tourism organization with offices in 10 or more different cities, and each prime geographical region (for example, the eight geographical regions in China; see Table 1) has at least one tourism service organization. Tourism organizations that met the data criteria were selected from the rankings of major tourism organizations in different service sectors (based on queries on three major tourism website platforms in China, including https://www.ctrip.com/, https://www.qunar.com/, and https://m.lvmama.com/). Another important criterion (that was purely practical) was whether sufficient information could be found on the home page of the organization's website. Therefore, this paper selected 38 tourism organizations according to the actual situation of tourism services in China (Table 2).

**Table 2.** List of the 38 representative tourism service organizations.

| Service Organization Types | Organization Name | Number |
|---|---|---|
| Famous hotels at home and abroad | **International famous hotel:**<br>Hilton Hotels Corporation, Intercontinental Hotels Group,<br>Marriott International, Inc. Hotels,<br>Harbor Plaza Hotels and Resorts Hotels,<br>Best Western International, CENDANT Corporation,<br>Kempinski Hotel, Starwood Hotels and Resorts Worldwide,<br>ACCOR, Shangri-La | 10 |
| | **Famous Hotels in China:**<br>Vienna Hotel, Jinjiang Inn, Hanting Hotel,<br>BTG Homeinns Hotels, Seven Days Hotel,<br>Green Tree Hospitality Group Hotel,<br>All Seasons Hotel, Atour Hotel,<br>Lavande Hotels, Super 8 Hotel | 10 |
| Air tourism passenger transport | Air China,<br>China Southern Airlines, China Eastern Airlines,<br>Hainan Airlines, Eva Airways, Shenzhen Airlines,<br>Juneyao Airlines, Spring Airlines, China Airlines,<br>Cathay Dragon Airlines, Cathay Pacific Airways,<br>Xiamen Air, Air Macao | 13 |
| Tourism consulting service agencies | China International Travel Service (CITS),<br>China Travel Service, China Comfort Travel Headquarter,<br>China CYTS Tours Holding Co., Ltd. (CYTS),<br>Citic Travel Co., Ltd. | 5 |

Here, 38 advanced tourism organizations were identified in three sectors, namely, 20 in hotel hospitality, 13 in aviation tourism, and 5 in tourism consulting service agencies. Sixty-three tourist cities were then used to create some measurement data for the 38 service organizations for a regional multivariate analysis of service-oriented tourist cities [25]. These efforts successfully stimulated a new data collection exercise for the tourism relationships of the 63 cities and 38 representative advanced tourism service organizations.

*3.4. Limitations of Cluster Analysis and Fuzzy c-Means Clustering Analysis*

Cluster analysis is one of the most popular empirical analysis techniques used to study a data matrix of a very large number of tourist cities [34]; clustering has a rich history in other disciplines, such as geology, geography, and marketing [35]. The field of spatial analysis of point patterns is related to cluster analysis [9]. Cluster analysis has been widely used in various interdisciplinary fields. It has been continuously improved, and many taxonomies of methods have been derived. Figure 1 is the summary of the traditional cluster analysis taxonomy. However, this traditional cluster analysis method is full of various problems in its practical application [36]. These problems are detailed below.

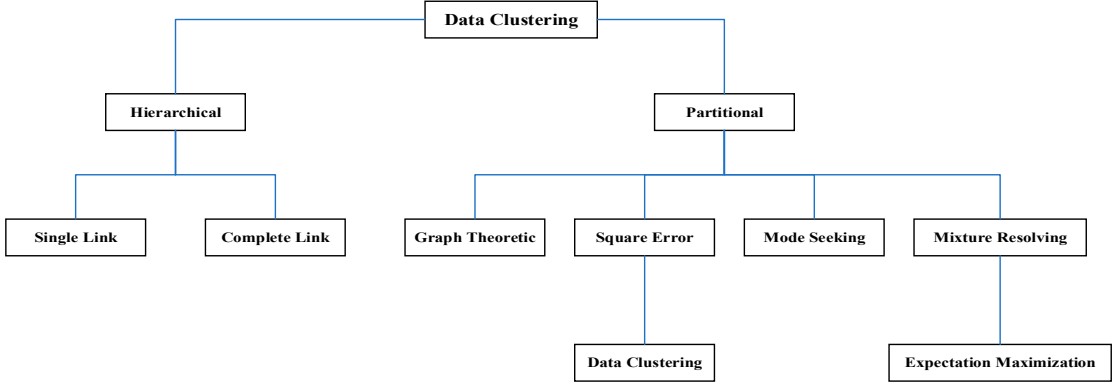

**Figure 1.** A taxonomy of clustering approaches (Jain, Anil K., 1999) [34].

First, P. J. Taylor et al. pointed out that the connected network of various tourist cities does not show a simple and absolute hierarchical structure of cities [6]. In fact, the connections between different tourist cities are a complex network rather than a simple hierarchical structure [32]. This is an empirical support for the opinion in Taylor's paper; he believes that the clear patterns provided by traditional clustering algorithms are unlikely to provide an accurate and sensitive specification in a complex city network [37].

Second, Chris Skelcher believes that functional patterns and hybrid hierarchy increase the complexity of the network [38]. When the networks of various scales are gathered together in a large region, the result is a complex network structure, showing multiple regional patterns and hierarchies. Therefore, the exploration of a service-oriented tourist city network should include the evaluation of the hierarchical structure of the overall region and the intertwining of regional patterns. Similarly, traditional crisp cluster classification analysis is unlikely to provide sensitive specifications for the scrambling patterns [39].

Third, Peter Taylor found that the previous classification evaluation is limited to the medium and upper rungs of the urban hierarchy system [40]. The main reason for this is that the outer areas of the urban network will be classified based on sparse data, so any classification based on the data will be vague [41]. Therefore, minor changes in sparse data usually yield completely different classification results, while, in theory, mutually exclusive cluster classifications will not be unbiased.

Traditional clustering approaches generate partitions; in a partition, each pattern belongs to one and only one cluster. Hence, the clusters in a hard clustering are disjointed. Fuzzy clustering extends this notion to associate each pattern with every cluster using a membership function. The output of such algorithms is a clustering, but not a partition. Fuzzy cluster theory was initially applied to clustering by Enrique (1969) [30]. The most popular fuzzy clustering algorithm is the fuzzy c-means (FCM) algorithm, which can ideally overcome the problems in traditional cluster analysis. The membership affiliation functions of FCM designed by Bezdek (1981) [35] are the most important advantage in fuzzy clustering; different choices include those based on similarity decomposition and centroids of clusters. Fuzzy c-means clustering is an advanced mathematical analysis method; it needs to be programmed with the help of the MATLAB software, in which packages are carried out. This analysis method can provide a comprehensive description of different types of tourist cities in the region, not just focusing on the medium and upper ranks of the urban hierarchy system, and it can show a comprehensive and non-absolute hierarchical urban structure with certain boundaries. Now, this research team will give a high-level partitional fuzzy clustering algorithm below [35].

In fuzzy clustering, every cluster is a fuzzy set of all the patterns. Figure 2 illustrates this idea. The rectangles enclose two clusters in the data: C1 = {1,2,3,4} and C2 = {5,6,7,8}. A fuzzy clustering algorithm might produce the two fuzzy clusters F1 and F2, depicted by ellipses. The patterns will have membership values in [0,1] for each cluster. For example, fuzzy clusters F1 and F2 could be compactly described as:

F1 = {(1,0.85), (2,0.80), (3,0.75), (4,0.65), (5,0.50), (6,0.35), (7,0.00), (8,0.00)};

F2 = {(1,0.00), (2,0.00), (3,0.35), (4,0.50), (5,0.65), (6,0.75), (7,0.80), (8,0.85)}.

To show the data structures more comprehensively, the crisp separation of clusters was replaced, defined by $\gamma_{it} \in [0,1]$ for all i = 1, 2, 3, ..., m and t = 1, 2, 3, ..., T, by a fuzzy notion, and $\gamma_{it}$ is the crisp membership of a city; m is the number of cities that need to be classified (63 cities in this paper); T is the number of clusters; and the ordered pairs (i, $\gamma_{it}$) represent the membership affiliation of city i in cluster t (ranging from 0 to 1) [9]. Larger membership affiliation values indicate higher confidence in the assignment of the city to the specific cluster. An elastic clustering can be obtained from a fuzzy partition by thresholding their membership affiliation values.

Based on dividing the membership affiliation value of different cities in the cluster into different numerical ranges, this also can reflect some cluster nuclei, singular members, hybrid members,

and near-isolated cities in the classification, which is closer to reality: The first type is a nuclear city, that is, the core of the cluster is composed of cities with an affiliation degree above 0.8; the second type is a singular city, and this is also true for members of a cluster with an affiliation between 0.3 and 0.8 that also have no important membership with other clusters; the third type refers to hybrid cities, which are members of a cluster that share membership with another cluster; the fourth type, nearly isolated cities, refers to cities that do not belong to a cluster (because the affiliation is less than 0.3), but have the highest membership affiliation with the given cluster.

The fuzzy clustering analysis of the classification calculates the hierarchies of membership of various tourist cities instead of providing information about membership alone [9]. This approach can easily reflect the expected complexity of intertwined and multiple profiles in any tourism connection classification because it is reflected by the hybrid membership relationships in various clusters. In addition, previous research results have indicated that the sensitivity of fuzzy clustering algorithms can simultaneously evaluate hierarchical tendencies and main city connection ranges [42]. Finally, the minor changes in the data will be reflected by the results of the city classification, thus reducing the occurrence of the bias problem. However, based on the fact that the classification of a few tourism organizations will be affected by the bias, this paper has limited the analysis to cases of representative tourism organizations with frequent services and has selected 38 advanced tourism service organizations with certain popularity in order to avoid such a sparse section in the matrix. Here, this produced a matrix dataset containing the distribution of 38 advanced tourism organizations in 63 tourist cities. Choosing a different number of clusters would yield significantly different results. Therefore, in the analysis of city classification, one should try their best to make the difference between the members of the cluster smaller and make the difference between different clusters as large as possible, and one should take these factors as the principle of city classification.

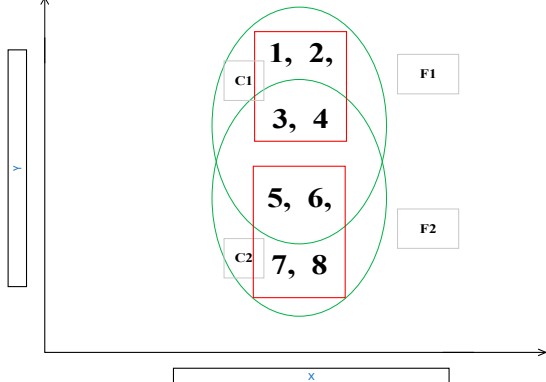

**Figure 2.** Fuzzy c-means clustering analysis (Jain, Anil K., 1999) [34].

### 3.5. Connectivity of the Urban Tourism Network

In this paper, the traditional clustering algorithm is applied to the relationship matrix generated based on the connections between various cities, and then the mutually exclusive city clustering is calculated according to the "tourism service value" of the service-oriented tourist city in the network.

The advantage of the accurate specification of connectivity in a network is that network analysis technology can be used. Using basic elementary network analysis, the most basic measure of a city is its connectivity in relation to all other cities in the matrix. According to Derudder [9], the interlocking network is defined as follows:

$$\lambda_{\text{ab},j} = V_{aj} \times V_{bj}, \tag{1}$$

where $\lambda_{ab,j}$ is the element of interlock connection between city a and city b in terms of tourism organization *j* defined in terms of matrix V, a tourism service value of a tourism organization in a city. These connections can be aggregated into intercity interlocking links:

$$\lambda_{ab} = \sum_{j=1}^{n} \lambda_{ab,j}, \tag{2}$$

where *j* = 1, 2, 3, ... , *n*, and *n* is the total number of tourism organizations. Every city has such an interlocking connection with other cities. All the internal connections of a city are aggregated to form the regional tourism network connectivity (C) of the city:

$$C_a = \sum_{i=1}^{m} \lambda_{ai}, \tag{3}$$

where, for city a across all cities in matrix V, *i* = 1, 2, 3, ... , *m*, and *m* is the total number of cities.

The limiting situation is that a city does not share tourism organizations with any other cities, so all these basic connections are 0 and it has no connectivity. In fact, in the case of large datasets, the connectivity of the urban tourism network may be quite large. To make them easy to manage in the following use, express city connectivity is expressed as the ratio of the largest connectivity calculated in the data, thus creating a scale from 0 to 1. These scores are used below to represent the hierarchical tendencies in the analysis.

## 4. Results: Hierarchical Tendencies and Regional Patterns of Tourist Cities

Through multiple debugging passes of fuzzy clustering analysis, this work focused on the result of T = 8, which is an ideal choice after evaluating several solutions of different clusters. Hierarchical tendencies were found in the eight clusters, and the regional patterns of the service-oriented tourist city network featured broad diversity. The connectivity difference between different clusters was obvious, which provides an ideal insightful interpretation here. In order to clarify the argument, the results of this paper will be expressed in a simplified form, and the "nucleus" and "hybrid" members in the cluster will be identified.

The new complex hierarchical tendencies and regional geographical patterns of tourist cities are shown in Table 3 and Figure 3. Table 3 highlights the hierarchical tendency differences in the results of city classification, and it can be clearly seen that there are obvious differences in the average connectivity between different clusters. Table 3 also shows the number of cities that contained hybrid members and the most typical cities in each cluster. These hierarchical tendency differences of average connectivity were used to represent the four arenas in turn to represent the hierarchical tendency distributions around cluster I, which is the most important in terms of city connectivity. The most central cluster I can be called the "center" of the regional tourism network in China, and the reason can be clearly expressed by the intuitive distribution in Figure 3. The divisions of the arena clusters are mainly determined by their status and function regarding the regional tourism connections. According to the fuzzy classification in Figure 3, the tourist cities in the regions have an obvious "center–edge" geographical structure.

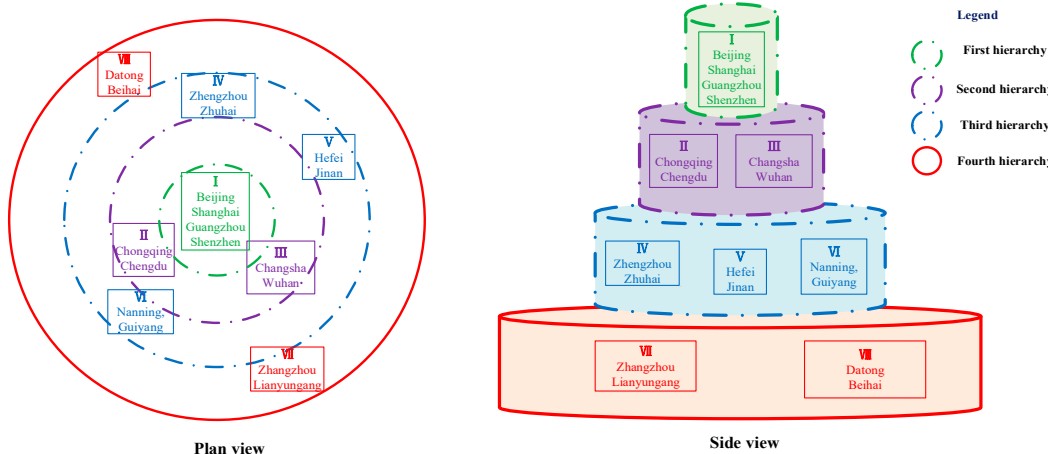

**Figure 3.** Tourism arenas of the service-oriented tourist city connection network. Note: Some typical cities in different clusters are listed in the text box. The dotted line between different hierarchies indicates that some hybrid cities can cross these boundaries.

The regional patterns of the city hierarchy in the results are reflected in different arenas in Figure 3. Different arena clusters are depicted as their respective banded areas around the center. In addition, most of their cities are roughly located in their geographical locations, which also proves the first law of geography, i.e., "the closer the cities are in spacing, the more likely they are to be connected." Figure 3 shows the hierarchical tendency distributions centered on Beijing, Shanghai, Guangzhou, and Shenzhen, which represent a large-scale regional structure across eight geographical regions in China. Eight clusters are distributed in four arenas with different tourism functions here, in addition to the nuclear cities in the first hierarchical arena. The other three arenas also have some nuclear cities with strong regional functions, and there are some relatively independent singular cities in a small scope, some hybrid cities with strong cross-regional influence, and a small number of nearly isolated cities in remote areas. As a whole, these four arenas have different tourism functions and represent different tourism chains, meaning that these four arenas have relatively clear regional characteristics. Therefore, the results of these regional patterns reflect the strengths of the city hierarchical tendencies.

**Table 3.** Bands of arenas in the service-oriented tourist city network.

| Cluster Arena | Service-Oriented Tourist City Band | Average Connectivity | No. of Members [1] | Typical Cities |
|---|---|---|---|---|
| I | First hierarchy | 0.958 | 4 | Beijing, Shanghai, Guangzhou, Shenzhen |
| II | Second hierarchy | 0.762 | 6(1) | Chongqing, Chengdu |
| III | Second hierarchy | 0.650 | 8(3) | Changsha, Wuhan, Hongkong, Taipei |
| IV | Third hierarchy | 0.541 | 14(3) | Zhengzhou, Zhuhai, Macao |
| V | Third hierarchy | 0.427 | 10(1) | Hefei, Jinan |
| VI | Third hierarchy | 0.325 | 12(2) | Nanning, Guiyang |
| VII | Fourth hierarchy | 0.234 | 11(5) | Zhangzhou, Lianyungang |
| VIII | Fourth hierarchy | 0.178 | 7(3) | Datong, Beihai |

Note: Memberships of clusters are defined here as an affiliation of 0.3 and above; hybrid cities with membership of other clusters are featured in brackets.

Further interpretation of these results requires detailed analysis of the content of the different arenas. According to the above four types of cities in the clusters, the cities in each cluster or arena are defined as four types of tourist cities: The nuclear city, the singular city, the hybrid city, and the nearly isolated city. From the above discussions, this work should focus on the regional patterns, and the analysis and discussion are ordered here through the differences of hierarchical tendencies.

### 4.1. The Leading Central Service-Oriented Tourist City and the First Hierarchy

Table 4 exhibits four types of cities in the first hierarchical arenas. The distribution of the first set of hierarchical cities is very simple, where it consists of four nuclear cities and nothing else.

**Table 4.** The leading central service-oriented tourist cities and the first hierarchy.

| City Type | I |
|---|---|
| Cluster nuclei | Beijing, Shanghai, Guangzhou, Shenzhen |
| Singular members | |
| Hybrid members | |
| Nearly isolated cities | |

Beyond the cluster nuclei, there are no singular members, hybrid members, or nearly isolated cities in the first hierarchical arenas. The arena at this hierarchy is composed of some important central tourist cities with international tourism influence. From the list of these first hierarchical cities, their connectivity degrees are all very high (Table 3). They are also completely independent and characteristic tourism service complexes. These factors are why the first hierarchical arena is designated as the leading center of the network and is the pivot point of tourism connection between other lower hierarchical cities (Figure 3).

These four cities in the first hierarchical arena are the leading service-oriented tourist cities that have specific relationships amongst other cities of different hierarchies. Beijing is China's political, cultural, educational, and traffic center, and these urban functions attract a wide range of tourists from home and abroad. Shanghai is China's economic, financial, international trade, and logistics center, and a lot of tourism demands there will be due to economic and trade needs. Guangzhou has been the commercial capital of China for thousands of years, and the Guangzhou Export Commodities Fair and other business activities attract various people from home and abroad to travel there. Shenzhen is the center of China's technological innovation, which attracts many foreigners for business tourism. It can be seen that the urban functions of the four cities are oriented to international demands and can serve a wide range of both international and domestic tourists. At the same time, their service characteristics are significantly different. The first hierarchical arena is a cross-regional service scope, connecting most other cities in China, including tourist cities with different service scales in eight geographical regions of China. However, the compositions of the main domestic tourist markets served by the four central cities are different. Beijing has more tourists that come from northern China. Shanghai has more tourists that come from eastern China and Taiwanese tourists. The group of tourists in Guangzhou are generally from southern and southwest China. The main tourists in Shenzhen are tourists from southern China, Hong Kong, and Macao. However, Sun has suggested that these differences in domestic tourist origins are more likely due to different geographical locations and, hence, different travel distances [43].

### 4.2. The Major Regional Service-Oriented Tourist City and the Second Hierarchy

The second hierarchical arena (Table 5) includes some important clusters of major regional tourist cities. There are two classic clusters in the second hierarchy: Cluster II is a distinctive cluster that includes all the important cities of western China (including northwest China and southwest China) not in the first hierarchical arena; cluster III is a cluster mainly composed of emerging tourist cities, including many important cities with a fast-growing economy developed by service-oriented tourist cities not in the first hierarchical arena.

**Table 5.** The major regional service-oriented tourist cities and the second hierarchy.

| City Type | Arena II | Arena III |
|---|---|---|
| Cluster nuclei | Chongqing<br>Chengdu<br>Xi'an<br>Nanjing<br>Hangzhou | Changsha<br>Wuhan<br>Qingdao |
| Singular members | | Sanya<br>Xiamen |
| Hybrid members | Hong Kong > III | Hong Kong > II<br>Taipei > III<br>Suzhou > III |
| Nearly isolated cities | | |

Arena II has no singular member, but arena III includes two coastal city cities as singular members, and the two cities have unique tourism characteristics in China. Xiamen is the coastal tourist city that is representative of a subtropical maritime monsoon climate. Sanya is the representative city of coastal tourism in the tropical monsoon climate region. They are currently hard to replace in China, and they serve national tourists, but they are geographically isolated from the central region and are not surrounded by strong tourism competitors and partners [44]. Arenas II and III both have important hybrid members. Arena II has a hybrid member, Hong Kong, which is connected with arena III. Hong Kong is a world-famous international metropolis, although, in recent years, due to the rapid development of the mainland cities, its comparative advantage has fallen a little, but its function of finance and trade is still an obvious advantage compared to the leading central service-oriented tourist cities. It still has frequent tourism connections with many important service-oriented tourist cities in China, so it is easy to understand why it is a hybrid city. Arena III has three hybrid members, where, in addition to Hong Kong, Taipei and Suzhou are also in the list. As a metropolis, Taipei has close tourist connections with important cities such as Shanghai, Nanjing, Xiamen, and Hangzhou. Suzhou is a city with extremely rich tourism resources, but it is obviously affected by both Shanghai and Nanjing. It is closer to Shanghai in terms of the tourism economy, but, in fact, it is directly affected by Nanjing in terms of the administrative jurisdiction.

In a word, all the cities of arenas II and III have independent and complete urban tourism functions. Although they are under the influence of the first hierarchical cities of the urban tourism connection network, they all have a stable tourism connection with other small- and medium-sized tourist cities within their own region, so these cities are typical regional tourist cities.

*4.3. The Nodal Service-Oriented Tourist City and the Third Hierarchy*

The service-oriented tourist cities of the third hierarchical arena (Table 6) have important node functions attached to the important tourist cities in the upper hierarchy and serve the tourist cities in the lower hierarchy.

There are three classic clusters in the third hierarchy: Cluster IV includes some provincial capital cities and economically developed cities that are not in the first and second hierarchical arenas; cluster V is a distinctive cluster that mainly includes some provincial capital cities with a medium economic level, as well as some specialized tourist cities, such as Guilin and Huangshan. Cluster VI is mainly composed of some traditional tourist cities like Qinhuangdao, Luoyang, Zhangjiajie, and other important city nodes of the Belt and Road Initiative, such as Quanzhou and Nanning. In addition, cluster VI includes three important provincial capital cities in northwest China as singular members. They are all important node cities in the vast northwest region, and the transportation choices for tourists to the northwest region will give priority to these three cities as transit cities. However, the geographical distance between them is far, and the connections between them are not frequent compared to between developed cities, but they are still important node cities within the scope of each

province. The three clusters in the third hierarchical arena have zero nearly isolated members, but they all have important hybrid members. These clusters share the unique structure of some tourist cities with other clusters, which is a typical feature of the third hierarchical arena. In cluster IV, besides Taipei and Suzhou, Macao also belongs to the hybrid city group. Macao is a special administrative region of China, protected by preferential policies for tourism. As an important city for gaming tourism, Macao attracts a very large number of tourists from other cities in China. Cluster V has one hybrid member, Macao, which is connected by serving many tourists from various cities of this cluster. Cluster VI has two hybrid members, Weihai and Yantai, which belong to the developed cities along the coast of Shandong Province. They also belong to a temperate monsoon climate and are adjacent to the Liaodong Peninsula of China, and they serve tourists from northern China, northeast China, and eastern China.

**Table 6.** The nodal service-oriented tourist cities and the third hierarchy.

| City Type | IV | V | VI |
|---|---|---|---|
| Cluster nuclei | Zhengzhou Zhuhai Dalian Tianjin Ningbo Wuxi Nanchang Kunming Shenyang Haikou Harbin | Hefei Jinan Guilin Fuzhou Shijiazhuang Hohhot Changchun Taiyuan Huangshan | Nanning Guiyang Quanzhou Qinhuangdao Luoyang Nantong Zhangjiajie |
| Singular members | | | Lanzhou Urumqi Xining |
| Hybrid members | Taipei > III Suzhou > III Macao > V | Macao > IV | Weihai > VII Yantai > VII |
| Nearly isolated cities | | | |

## 4.4. The Service-Oriented Tourist City on the Edge and the Fourth Hierarchy

The fourth hierarchical arena generally includes tourist cities with low tourism popularity in the current period (Table 7). These cities are classified as "edge cities" in China's urban tourism service network. This "edge" is not because their geographical location is in a remote place, but because most of them do not fully enjoy the development opportunities brought about by network connection and are in an edge position of the network. This reflects the low and relatively isolated positions of some ordinary tourist cities in the service-oriented tourist city network. However, this situation does not prove that they are not involved in process of tourism regionalization in the urban tourism network. However, their connectivity in the network of service-oriented tourist cities in China is not high, and some cities do not have a high degree of affiliation in the corresponding cluster and have fewer tourism connections with other cities, so they belong to the "nearly isolated" group of cities, such as Lasa and Jilin, in this case. Moreover, there are many hybrid members. Zhangzhou, Lianyungang, Weihai, Yantai, and Zhongshan of the fourth hierarchical arena, as hybrid cities, may be dissatisfied with the development level of tourism services, so they try to establish connections with cities in other clusters in order to improve their tourism services in order to change their edge position.

**Table 7.** The edge service-oriented tourist cities and the fourth hierarchy.

| City Type | VII | VIII |
|---|---|---|
| Cluster nuclei | Yinchuan<br>Chengde<br>Shantou<br>Zhanjiang | Datong<br>Beihai<br>Jiujiang |
| Singular members | Wenzhou | |
| Hybrid members | Zhangzhou > VIII<br>Lianyungang > VIII<br>Weihai > VI<br>Yantai > VI<br>Zhongshan > VIII | Zhongshan > VII<br>Zhangzhou > VII<br>Lianyungang > VII |
| Nearly isolated cities | Lasa | Jilin |

The main reasons why cities at this level have not established good tourism service connections with other cities are given as follows.

First, they lack a competitive tourist attraction. Some cities in this hierarchy have some traditional tourism resources, such as sightseeing and worship locations, but they lack high-level tourism resources with strong attraction in comparison to their current tourism services.

Second, the cities' functions are relatively backwards and singular, and outsiders lack the necessity of entering the city. Here, it was found that the travel services of these lower-level arena members may be more regional, but their service areas are not large. The cluster VII nuclei, including Yinchuan, Chengde, Shantou, and Zhanjiang, and the cluster VIII nuclei, including Datong, Beihai, and Jiujiang, have their own relatively stable tourist service groups, and other stronger tourist cities have not formed fierce competition with the tourism services of these nuclear cities due to the cost resulting from the spatial distance [45] or because of a lack of competitive tourism resources.

Finally, Wenzhou, as an independent service-oriented tourist city, exists in the coastal area in the southern area of Zhejiang Province. There is no strong service-oriented tourist city within 200 km of the city. The city mainly meets the demands of surrounding areas for tourism services. Thus, it is classified as a singular city. Generally speaking, the fourth hierarchy of tourist cities, as the basis of the above three hierarchical arenas, has an irreplaceable role, but this edge position may change with the construction of tourism attractions, traffic improvement, and changes in city development strategies [6].

*4.5. Arena Gaps of Regional Geographies*

These findings come from the analysis of the regional tourist city network in China. However, famous "center–edge" theories point out the identification of the city network arena. Next, this work analyzes the regionalization of contemporary tourism flows in terms of promoting closer connections within cities in a specific region. This may impinge upon the long-established regional "city network system", which is the traditional focus of tourism geographers in terms of analyzing the relationships between various cities. Figure 3 depicts the "center–edge" structure in this network of China, which gives us some special information.

First, our research team can compare the location of the arena centered on China's three major economically developed regions. The major cities of eastern China, centered on Shanghai, are mainly in two hierarchical arena bands (Shanghai is in the first hierarchy and Nanjing and Hangzhou are in the second hierarchy). The important cities of southern China, centered on Guangzhou and Hong Kong, are mainly distributed in three hierarchical arena bands (Guangzhou and Shenzhen are in the first hierarchy, Hong Kong is in the second hierarchy, and Macao is in the third hierarchy). The important cities in northern China, centered on Beijing, are mainly in two hierarchical arena bands (Beijing is in the first hierarchy and Tianjin and Shijiazhuang are in the third hierarchy), where the regional city system has no cities in the second hierarchy. These results clearly show that, compared with the

"vertical" city relationship of the network of tourist cities in northern China, the networks of the other two regions have a more reasonable "horizontal" city relationship. This good cooperation shows that the tourism service connections between various cities in these two regions have enabled most cities of the two regions to develop healthily and to fully enjoy the benefits of regional service connections. Although these results are definitely not surprising, they are closely related to the connection of tourism regionalization. Beijing has adopted absorptive policies regarding the surrounding cities in urban agglomerations, such that the high-quality resources of urban service functions, such as transportation, exhibition, education, medical treatment, and high-end shopping, in the surrounding areas are concentrated in the core cities. The other two regions pay attention to the spatial distribution of urban service functions in order to form complementary service resources among various cities. Clearly, the development strategy of urban tourism services is of vital important for the relationship between the service function development of each city relating to the process of tourism globalization.

Second, in addition to China's three most important urban agglomerations, southwestern China, represented by Chongqing and Chengdu, and central China, represented by Changsha and Wuhan, both have a similar pattern for the regional tourism service arena to that of the urban network system of southern China. Both of them are represented as a regional tourism service network model centered around a binary star. Chongqing and Chengdu are nuclei of cluster II in the tourism network of southern China. Changsha and Wuhan are nuclei of cluster III in the tourism network of central China. In view of the fact that there are many tourist cities in southern and central China in higher hierarchical arenas, this shows that the two regions have also gradually formed regional tourism connection clusters. However, this fact is more interesting in terms of the distribution of the third and fourth hierarchical arenas: In the process of tourism globalization in southwestern China, there seems to be a relationship between very important cities, including Chongqing and Chengdu, and tourist cities, including Kunming and Guilin, in the southwestern region. There is a large gap in service development, and this gap in development is growing in the current state. In fact, Kunming and Guilin were developed cities in China's early tourism service. However, in the past decade or so, their tourism service competitiveness has experienced a relative decline. However, in the process of tourism service globalization in central China, the advantageous gap between the two nuclear cities and other tourist cities, such as Zhengzhou, Nanchang, and Hefei, is narrowing. At present, most of the cities in this region are experiencing a rapid upward trend, which means that, under the incentive of tourism globalization, most cities in the tourism service arena in the service network of central China are trying to "move up in the network" and create a new central China arena at a higher hierarchy level. This shows, on the other hand, that most of the cities in central China can share the benefits brought by arena connection and cooperation. On the whole, the two regions are similar in terms of their tourism service networks. Chongqing and Chengdu in southwest China are more developed than Changsha and Wuhan in central China. However, the overall development level of central China is more ideal. The reason for this gap is that the population size and urban functions of the nuclear cities in southwest China are too concentrated, while most cities in central China have a large population size, and they are gradually improving their regional transportation systems and commercial tourism service markets.

Thirdly, the most attractive achievement of this paper is that it also shines light on the former tourist cities that were easily neglected in the city network. The important viewpoint is that the urban competitive arenas of tourism service in northwestern and northeastern China are well represented in the network beyond the main service regions. Taking northwestern China as an example, Xi'an and Hohhot are relatively important representatives of tourism services. Both Dalian and Shenyang have gathered around a strong performance within northeastern China. At the same time, Hong Kong, Taipei, and Macao are hybrid cities, indicating that their service network systems are not limited to a single cluster, and, because of their extensive influence, they participate in multiple tourism service connections. It is worth mentioning that tourist cities such as Lasa and Jilin are in the network as nearly isolated members. In fact, they all have good tourism resources, such as the Tibetan cultural

landscape of Lasa and the magnificent landscape of the Songhua River in Jilin. The possible reasons for being nearly isolated members include Lasa being geographically far away from cities in relatively developed regions, such as eastern, southern, and northern China, and that the traffic accessibility of Lasa is also poor. Jilin is a traditional industrialized city; the local industry has declined, and its urban functions have not made significant relative progress in the past ten years.

## 5. Discussion

(i) From Table 3, it can be seen that the average connectivity of tourist cities with the first hierarchical tendency is as high as 0.958, which is much higher than other cities in other hierarchies, while the connectivity values between middle- and low-hierarchical-tendency tourist cities are not significantly different. Differently from the intuitive feeling of most people, with the improvement of transportation and communication, the network of tourist cities based on tourism services will gradually mature; one can further speculate that the connectivity advantage of high-hierarchical-tendency tourist cities in the network will continue to be maintained, and the connectivity between middle- to low-hierarchical-tendency cities will be improved as a whole, but the relative differences in connectivity between them will be reduced. The reason for this judgment is that the tourism service values in the network are reflected by tourists as the service objects, that is, these tourism services are provided to tourists and they also have hierarchy differences in terms of service consumption [46]. This viewpoint is very close to Bao Jigang's research; he thinks that high-end tourism services must exist in high-hierarchy tourist cities with corresponding service functions [47]. High-hierarchy tourist cities have service functions that cannot be replaced by other cities in a large region. For example, international tourism exhibition activities are generally selected in Guangzhou, Beijing, and other high-hierarchy service-oriented tourist cities, where other low-hierarchy cities may not have a complete set of high-quality accommodation, hotels, airplanes, translation and consulting services, and other institutions to support the tourism exhibition services. At the same time, the urban functions of middle- to low-hierarchical-tendency cities are generally easily replaced by surrounding cities, so the tourism service competition among them is more intense. This competitive situation is similar to the state of free competition, which will make the connectivity between them tend towards an average state. This conclusion is a good supplement to Friedman's research theory [5]. Therefore, it is not difficult for us to understand that the effective impact area of tourism services of cities can only include a small regional scope, and the lower the hierarchy of tourist cities, the smaller the effective impact of the regional scope. However, when the tourism service network becomes mature and perfect, high-hierarchy cities' tourism service scopes become larger because of the unique high-end urban functions among them, where their tourism competitions are, surprisingly, not fierce.

(ii) From the perspective of regional patterns, due to the huge geographical space of China, its tourism service network still shows a highly polycentric pattern, and high-hierarchy service-oriented tourist cities often play the role of central cities, with Beijing as the center in northern and northeastern China, Xi'an as the center in northwestern China, and Chongqing and Chengdu as double centers in southwestern China. Changsha and Wuhan are double centers in the central region of China, Shanghai is the center in eastern China, and Guangzhou and Shenzhen are double centers in southern China. Due to the construction of high-speed rail and air transport, the urban functions of these central cities will be strengthened within their regional scopes, which will further increase the tendencies of surrounding tourism services flowing to central cities and also improve their network connectivity in their respective regional scopes.

With the improvement of transport and information, edge tourist cities can take advantage of their own tourism characteristics to fulfill the secondary tourist service needs of the surrounding central cities and enjoy the positive externalities brought about by the development of the central cities. This will improve their own connectivity performance in the service network and promote their connectivity to be narrowed between middle- to low-hierarchical-tendency tourist cities [40].

Therefore, it can be seen that there will still be a "center–edge" regional distribution structure in China's overall tourism service network. With the maturity of the tourism service network, the positions of the central cities may be strengthened, while the edge cities may be reduced. The new "center–edge" model based on the tourism service network is a theoretical improvement for "center–edge" theory in terms of traditional geography. Unlike the work of Derudder et al. [9], this study has two novelties that lie in the "center–edge" theory as follows: First, the new "center–edge" model is a polycentric three-dimensional connectivity network; the network can be divided into four different hierarchical tendencies, and they also have four functional roles in the network. In the fuzzy c-means clustering analysis, we find that there are hybrid tourist cities, and the analysis found that different "center–edge" clusters or arenas also have the possibility of cross-group interaction, which is closer to the reality of tourism services. Second, the new "center–edge" model considers bidirectional flow, and the central cities (high-hierarchy cities) can provide tourism service products and tourists to edge cities (low-hierarchy cities), and edge cities can also provide tourism service products and tourists to high-hierarchy cities. However, the traditional "center–edge theory" often implies that edge cities provide agricultural products or developmental capital to central cities, while the central cities provide industrial products or services for the edge cities. This flow process cannot flow in two directions at the same time. Therefore, the new model in this paper enriches the application of "center–edge" theory in new research scenarios.

(iii) The superficial differences between high- and low-hierarchy cities lie in the sizes of cities, the number of tourists, and the levels of connectivity. Differently from the belief of U. Mans about the reasons for the difference in network node positions [13], this paper argues that the essence of this difference lies in the difference of urban tourism service functions: High-hierarchy cities have high-end urban tourism service functions that are not easily replaced, such as the entertainment of Disneyland, famous international high-end hotels, and international passenger airports, and there is no such service function in the surrounding cities, so the potential tourists in these large regions will choose these high-hierarchy tourist cities to meet their high-level service demands, and the high-hierarchy tourist cities also have most of the tourism service functions of ordinary cities. Therefore, high-hierarchy tourist cities have a wider range of urban tourism service functions that can meet the needs of tourists with different tourism service demands. Compared with these cities, middle–low-hierarchy service-oriented tourist cities are limited by their tourism service functions, and their effective tourism influence scopes are not always wide, so they may have a certain comparative advantage in small geographical regions due to proximity.

In the face of this situation, how should high-hierarchy tourist cities and middle–low-hierarchy tourist cities adopt appropriate methods for scientific development of the tourism service network?

Firstly, tourist cities with high hierarchical tendencies (central cities) need to constantly seek new urban service functions to maintain their advantages regarding their tourism service functions. What cannot be ignored is that the development of Chinese cities is obviously affected by administrative forces. For example, Shenzhen was still a small fishing village in 1980. Driven by the preferential policies brought about by administrative power, Shenzhen developed rapidly and became an important city that exists in the first hierarchy. Therefore, high-hierarchy cities need to participate in the formulation of tourism development planning at the national level, make use of transportation, and encourage communication, construction, and other means to reasonably layout their regional tourism service advantages so as to avoid the influence of administrative power on their core tourism service functions.

Secondly, tourist cities with middle hierarchical tendencies need to do a good job in the tourism convergence service between the central city and the edge city, actively absorb the tourists from the central city, and explore the tourism service functions with their own characteristics (such as their long histories and cultural characteristics, slow rhythm of life of the city, seafood cities, etc.). At the same time, these cities should strengthen contact with their edge cities, build a tourism service platform for them, and enhance the influence of regional tourism services with them.

Finally, tourist cities with low hierarchical tendencies (edge cities) need to improve their urban tourism service functions, form a more comprehensive service reception capacity, and expand the number of tourists received so as to increase tourist satisfaction. At the same time, these cities need to actively pay attention to the influence of the administrative power of Chinese cities and create their own tourism service characteristics with the help of tourism planning (such as healthcare locations, holiday resorts, natural landscape wonders, etc.) to gradually establish their own tourism images in the regional tourism scope.

## 6. Conclusions

Based on Manuel Castells' theory of the flow space network [2], this paper has enriched the theory of city position and classification in a flow space network using fuzzy c-means clustering analysis. In addition, from the GaWC's research [9], this paper has developed an empirical method to study the formation of tourist city networks. The method improves the spatial structure theory of tourism geography in a relatively complete large national region, and especially complements the "center–edge" theory.

The results here obviously reveal the differences between connectivity and geographical distribution from the data analysis, indicating the regional pattern characteristics and hierarchical tendencies' distinctions of cities in the network. These hierarchical tendencies and regional patterns prompt three conclusions.

Firstly, China's four most important international cities, as central structures, lead the whole tourism connection network, and the connectivity of each city is very high. As the first hierarchical group of cities, these four cities have formed a unique tourism arena. They all have their own unique urban functions, and their tourism services are not easily replaced by other cities. In terms of their geographical distribution, they occupy the northern, eastern, and southern regions of China. This spatial layout can effectively connect them to more low-hierarchy cities in the geographical space. At the same time, there is an obvious phenomenon in the interaction between hierarchical tendencies and regional patterns, that is, clusters with low average connectivity are more restricted by regional forces in special regions, and they have less influence on cities that are farther away. This geographical phenomenon further shows that a regional pattern is not only the location space of different clusters, but also represents the interaction results of various tourism arenas in the urban network with different hierarchical tendencies.

Secondly, the regional patterns show that eastern, southern, and northern China are the three most important arenas of the tourism service economy in China, but there are great differences in the configurations of these arena. In northern China, the configuration is different from those in eastern China and southern China in the following aspects: (1) There is a "vertical" structure of service connectivity in tourist cities, and there is no second-hierarchy service-oriented tourist city. Beijing's comparative advantages and scale of service connectivity are both far stronger than those of the surrounding tourist cities. (2) In the process of tourism service globalization, high-quality urban functions are excessively concentrated in the nuclear cities. In contrast, the tourist cities in eastern and southern China have better service connectivity on the whole, and their cities cover all arena hierarchies. The connectivity scale distribution structure of the tourist cities of the two regions tends to be "horizontal", and tourist cities in eastern China perform better. The two regions of southwestern and central China are similar in terms of their tourism service network. Chongqing and Chengdu in southwestern China are more developed than Changsha and Wuhan in central China, but the overall development level of central China is more ideal. At the same time, this paper highlights the former "previously easily neglected tourist cities" in the service-oriented tourist city network, and northwestern and northeastern China are well-represented in the network beyond the main service regions. Xi'an and Hohhot are relatively important representatives of the tourism services in northwestern China, and Dalian and Shenyang have gathered a strong performance within northeastern China.

Thirdly, this paper has attempted to improve the understanding of tourism regionalization by using a new and detailed description of a tourism connection network based on regional patterns. Instead of limiting "region forces" to several major "international cities", this work incorporated a very large number of various tourist cities into a single integrated region to construct an analysis framework for urban tourism connections. The geographical distribution of contemporary tourism service globalization in the regional scope is not an end-product, but an important part in a series of continuous processes. This means that as the connectivity of the service-oriented tourist city network intensifies, the gaps identified in the network will be filled up in the next few years. On the other hand, with the development of a regional growth pole at the national level, the regional service industry of urban tourism may be more and more concentrated in a few important tourist cities, and the gaps of the tourism connection network may widen. Due to the rapid development of China's urban tourism, these hierarchical tendencies and regional patterns of the future tourism network should also change. In the face of such unknown changes, city managers should have a scientific understanding of the contemporary urban tourism network; otherwise, they will not be able to evaluate these changes, nor can timely and correct updates be made to the strategies of urban tourism development in China.

**Funding:** General program of Hunan Natural Science Foundation (2020JJ4373). Title: Study on residents' livelihood changes and environmental behavior of ecological restoration tourist destinations under the policy of Fallow. The research team has received funding from the project for the research work of this paper.

**Acknowledgments:** The authors are thankful for the support of the research platform from Tourism Development and Planning Research Center, Sun Yat-Sen University. We would like to acknowledge the support from all of the participating students and the rural cadres. The authors are deeply grateful for the useful comments and suggestions from the editor and review experts.

**Conflicts of Interest:** The author declares no conflict of interest.

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
