# Peer review of "Regional Patterns and Hierarchical Tendencies: Analysis of the Network Connectivity of 63 Service-Oriented Tourist Cities in China"

_sustainability, doi:10.3390/su12166532_

Round 1
Reviewer 1 Report
The manuscript presents a complete and critical review of regional pattern and hierarchical tendencies of service-oriented tourist city network in China.
My suggestion to the authors:
The purpose of the research should be clearly defined (the same purpose should be written in the summary and at the end of the introduction). In the current version of the manuscript, the authors define the purpose of the research differently in several places (e.g. lines:39-40; 62-66; 189; 228-230).
Please read the article carefully and eliminate all unnecessary repetition of the text:
- The content of the introduction is repeated in the literature review, methodology and conclusions.
- The methodology is also too extensive and duplicates the same content (e.g. point 3.1 and point 3.3)
Lines 327-331 should be in the literature review
From the formal aspects the following corrections should be made:
- remove [backgroung] - line 9, [method] - line 12, [results] - line 15 and [conclusion] - line 19
- remowe "tourism connection network" - line 149
- manuscript need corrections of text editing
Author Response
Dear Reviewer 1:
Thank you for your understanding and support for the paper. Although the revision process of the paper is very hard, I have made progress in the revision. When the article is about to be revised, I would like to express my best wishes to you for these comments and suggestions!
Kind regards.
Sincerely yours,
Please see the attachment.
Author

Reviewer 2 Report
Dear Authors,
your research work falls within the objectives of the journal. However, the modest form of the English text, as well as the equally poor structure of the manuscript, do not allow to fully detect the originality of the work, although I detected that it could be original.
I think that for a possible publication the work should be improved by making important changes. First of all, English should be deeply revised in the form because the sentences are long and complicated to read and sometimes appear incomplete.
In addition, the structure of the text needs to be improved. In detail:
In the “introduction: the novelty of the study must be clearly highlighted and this part can be written well only after discussing the results obtained by comparing your results with other analogues in the literature in the appropriate section. The discussion section currently lacks such information. Furthermore, in Line 26: Castells’ theory: Please provide here a brief description of this theory or at least a relevant reference. 
Lines 52-54: this paragraph is not clear. Please improve English form.
In the literature review section: a great part of this section, should be included in the Methodology section because it explains the theoretical framework adopted in the study. In the literature section the authors should include the relevant literature that regards research study similar to that analysed in their research, highlighting any points of weakness that will be overcame by their research work. This is crucial to improve in the introduction section where the novelty of the research should be better described (see lines 65-67).
In the section methodology: 3 please move here part of the text reported in section 2, i.e., all the part which are used ad method for your study. Furthermore, since the methodology framework takes into account several steps described in the previous section, in order to improve the readability of the paper, the Authors should provide a flowchart to describe the different phase of the method. In the present form section 3 seems to describe a mere case study and it does not look like an original method to overcame limitations that should be well described in the introduction section.
Section 3.2 data production: line 275: Credible? Please clarify! What do you mean with credible?
lines 279-284. "0 means there is no distribution, 2 as the normal conditions, and 5 as the highest " Please, justify this sentence is not clear. Are any relevant studies that support this in literature?

Section 3.3 data collection: line 316: “Precise specifications guide our data collection process relatively smooth “. Please, clarify.
Section 5. Discussion and Conclusion: This section is more conclusion than discussion. It misses discussion with other research papers in order to highlight the novelty of this study. The authors cite only the paper that developed the method adopted. In the present form, this research is a case study.
Author Response
Response Letter
Dear Reviewer 2:
I would like to express our sincere appreciation for the reviewer’ valuable comments on our manuscript. I have carefully reviewed the suggestions and have revised the manuscript accordingly. My responses are given in a point-by-point manner below. Please find below for our responses to each comment in detail. I have highlighted all the revisions in the manuscript in red, and please refer to the manuscript for complete revisions.Thank you very much for all your help.
Best regards!
Modification Description:
- Your suggestions or comments are in Black Times New Roman characters;
- Red handwriting indicates that it needs to be deleted;
- The green handwriting indicates the adjusted statement;
- The position of the horizontal line indicated in the arc bracket indicates the position of the modification in the original text.
Point 1: First of all, English should be deeply revised in the form because the sentences are long and complicated to read and sometimes appear incomplete.
Response: Thank you for your comments. I have made corresponding modifications to your valuable suggestions. At the same time, I handed the revised article to the relevant agencies of Sustainability for language editing. The main adjustments are as follows:
First: the purpose of the research had be clearly defined, they had written at the end of the introduction.for example:
I have Delete “The main purpose of this study is to rectify this research limitation: We investigated a great quantity of representative tourist cities in specific regions. ” (lines:39-40)
I have Delete“In order to achieve research objectives, our empirical analysis concentrates on the latter type of research, understanding the service-oriented tourist city configuration within the relational data. ” (lines:228-230)
I have adjusted the sentence: “and we have tried to develop a method to analyze and measure the networks of service-oriented tourist city.” (line:189)
I have adjusted the sentence:
“Through our academic efforts, we analyze the network connectivity characteristics of tourist cities with various hierarchical tendencies from the perspective of overall geographical region, to improve the division and evaluation field of city connectivity network in tourism geography research. ” (lines:62-66)
Second,I have modified and adjusted some content of the introduction repeated in the literature review, methodology and conclusions.
For examples:
I re-checked the repeated contents of the introduction in the literature review, methodology and conclusions. The specific modifications are as follows:
“the representative tourist cities in the region construct the spatial form of network structure through tourism service connection. It is composed of three aspects” (lines:87-89)
The sentence is adjusted: “ the network structure constructed by tourism service connection of tourism city mainly includes three aspects ”.
“Through understanding the flow network structure of tourist cities with different hierarchical tendencies in a region can better formulate the tourism development strategies of different ranks of cities and promote the healthy development of regional tourism.” (lines:92-95) Deleted
“To analysis scientifically the hierarchical tendencies and regional patterns of various cities”(line:97)
Deleted
They are all relatively famous service-oriented tourist cities in China, and the cities are selected for their network connectivity is at least one twelfth network connectivity of the highest city according to the preliminary calculation. (lines:266-268) Deleted
”The analysis of cities should not be as many as possible, because as the size of the relational data matrix increases (i.e. more cities have been added.), it may become relatively "sparse" (many zero entries), which reduces the reliability of the analysis. So, these cities were selected for their network connectivity is at least one twelfth network connectivity of the highest city. ”(lines:339-342) Deleted
“In this paper, we supplement the previous exploratory analysis of the service-oriented tourist city network, the main objectives are: on the one hand, the fuzzy spatial dimensions behind the formation of tourist cities are cleared up; on the other hand, try to describe the geographical details of the network, in order to illustrate geographical space is an important factor in the formation of the network.”
(lines:630-634)
I have adjusted the sentence in So, the main research objectives are: on the one hand, the fuzzy spatial dimensions behind the formation of tourist cities are cleared up; on the other hand, try to describe the geographical details of the network, in order to illustrate geographical space is an important factor in the formation of the network. (in page 2, the last paragraph of the section “1. Introduction”)
The research specifies tourist cities as a city interlocking networks that we apply a regional holistic analysis of them in China.(lines:635-636) Deleted
Third, I have adjusted the expression of the methodology part as follows:
“In this empirical data collection, we are faced with the information of some tourism organizations is very detailed and the information of other tourism organizations is much less.” (lines:288-289)
I have adjusted the sentence:
“In the production of data, we find that some tourism organizations have very detailed information, while others have much less information.”
“ the query objects are the three kind of major tourism service sectors which include famous hotels, air passenger transport and tourism consulting service agencies.” (lines:323-325) Deleted
Point 2: In the “introduction: the novelty of the study must be clearly highlighted and this part can be written well only after discussing the results obtained by comparing your results with other analogues in the literature in the appropriate section. The discussion section currently lacks such information.
Response: Thank you for your advice. I re-checked the repeated contents of the introduction in the literature review, and write the discussion and conclusion separately:
In the disscusion:
the first part mainly discusses the author's judgment on the future development tendencies of tourism network model based on empirical discussion;
the second part discusses the novelty of the "center edge" structure of tourism service connection network;
the third part discusses the essential reasons for the difference of tourism service connectivity, and the reasonable development measures of different levels of tourism cities.
The contents of Discussion are as follows:
(i) From Table 3, it can be seen that the average connectivity of tourist cities at the first hierarchical tendency is as high as 0.958, which is much higher than other cities in other hierarchies, while the connectivity values between middle to low hierarchical tendency tourist cities are not significantly different. Different from the intuitive feeling of most people, with the improvement of transportation and communication, the network of tourist cities based on tourism services will gradually mature, where one can further speculate that the connectivity advantage of high hierarchical tendency tourist cities in the network will continue to be maintained and the connectivity between middle to low hierarchical tendency cities will be improved as a whole, but the relative differences in connectivity between them will be reduced. The reason for this judgment is that the tourism service values in the network are reflected by tourists as the service objects, that is, these tourism services are provided to tourists and they also have hierarchy differences in terms of service consumption [43]. This viewpoint is very close to Bao Jigang's research, He thinks that High-end tourism services must exist in high-hierarchy tourist cities with corresponding service functions [44]. High-hierarchy tourist cities have service functions which cannot be replaced by other cities in a large region. For example, international tourism exhibition activities are generally selected in Guangzhou, Beijing, and other high-hierarchy service-oriented tourist cities, where other low-hierarchy cities may not have a complete set of high-quality accommodation, hotels, airplanes, translation and consulting services, and other institutions to support the tourism exhibition services. At the same time, the urban functions of middle to low hierarchical tendency cities are generally easily replaced by surrounding cities, so the tourism service competition among them is more intense. This competitive situation is similar to the state of free competition, which will make the connectivity between them tend towards an average state. This conclusion is a good supplement to Friedmann's research theory[5]. Therefore, it is not difficult for us to understand that the effective impact area of tourism services of cities can only include a small regional scope, and the lower the hierarchy of tourist cities, the smaller the effective impact of the regional scope. However, when the tourism service network becomes mature and perfect, high- hierarchy cities’ tourism service scopes become larger because of the unique high-end urban functions among them, where their tourism competitions are surprisingly not fierce.
(ii) From the perspective of regional patterns, due to the huge geographical space of China, its tourism service network still shows a highly polycentric pattern, and high-hierarchy service-oriented tourist cities often play the role of central cities, with Beijing as the center in northern and northeastern China, Xi'an as the center in northwestern China, and Chongqing and Chengdu as double centers in southwestern China. Changsha and Wuhan are double centers in the central region of China, Shanghai is the center in eastern China, and Guangzhou and Shenzhen are double centers in southern China. Due to the construction of high-speed rail and air transport, the urban functions of these central cities will be strengthened within their regional scopes, which will further increase the tendencies of surrounding tourism services flowing to central cities and also improve their network connectivity in their respective regional scopes.
With the improvement of transport and information, edge tourist cities can take advantage of their own tourism characteristics to fulfill the secondary tourist service needs of the surrounding central cities and enjoy the positive externalities brought about by the development of the central cities. This will improve their own connectivity performance in the service network and promote their connectivity to be narrowed between middle to low hierarchical tendency tourist cities [35].
Therefore, it can be seen that there will still be a “center-edge” regional distribution structure in China's overall tourism service network. With the maturity of the tourism service network, the positions of the central cities may be strengthened, while the edge cities may be reduced. The new “center-edge” model based on the tourism service network is a theoretical improvement for “center-edge” theory in terms of traditional geography. Unlike the work of Derudder et al.[28], I find that this study has two novelties lies in the "center-edge" theory as following: First, the new “center-edge” model is a polycentric three-dimensional connectivity network, the network can be divided into four different hierarchical tendencies, and they also have four functional roles in the network. In the fuzzy c-means clustering analysis, we find that there are hybrid tourist cities were found here, and the analysis found that different “center-edge” clusters or arenas also have the possibility of cross-group interaction, which is closer to the reality of tourism services. Second, the new "center-edge" model considers bidirectional flow, and the central city (high-hierarchy cities) can provide tourism service products and tourists to edge cities (low-hierarchy cities), and edge cities can also provide tourism service products and tourists to high-hierarchy cities. However, the traditional “center-edge theory” often thinks that edge cities provide agricultural products or development capital to central cities, while the central cities provide industrial products or services for the edge cities. This flow process cannot flow in two directions at the same time. Therefore, the new model in this paper enriches the application of “center-edge” theory in new research scenarios.
(iii) The superficial differences between high- and low-hierarchy cities lie in the sizes of cities, the number of tourists, and the levels of connectivity. Different from Mans U believes that the reasons for the difference in network node position[11], this paper argues that the essence of this difference lies in the difference of urban tourism service functions: High hierarchy cities have high-end urban tourism service functions that are not easily replaced, such as the entertainment of Disneyland, international high-end famous hotels, and international passenger airports, and there is no such service function in the surrounding cities, so the potential tourists in this large region will choose this high-hierarchy tourist cities to meet their high-level service demands, and the high-hierarchy tourist cities also have most of the tourism service functions of ordinary cities. Therefore, high-hierarchy tourist cities have a wider range of urban tourism service functions which can meet the needs of tourists with different tourism service demands. Compared with these cities, middle-low hierarchy service-oriented tourist cities are limited by their tourism service functions, and their effective tourism influence scopes are always not wide, so they may have a certain comparative advantage in a small geographical region due to proximity.
In the face of these situation, how should high-hierarchy tourist cities and middle-low hierarchy tourist cities adopt appropriate methods for scientific development of the tourism service network?
Firstly, tourist cities with high hierarchical tendencies (central cities) need to constantly seek new urban service functions to maintain their advantages regarding their tourism service functions. What cannot be ignored is that the development of Chinese cities is obviously affected by administrative forces. For example, Shenzhen was still a small fishing village in 1980. Driven by the preferential policies brought about by administrative power, Shenzhen developed rapidly and became an important city that exists in the first hierarchy. Therefore, high-hierarchy cities need to participate in the formulation of tourism development planning at the national level, make use of transportation, and encourage communication, construction, and other means to reasonably layout their regional tourism service advantages, so as to avoid the influence of administrative power on their core tourism service functions.
Secondly, tourist cities with middle hierarchical tendencies need to do a good job in the tourism convergence service between the central city and the edge city, actively absorb the tourists from the central city, and explore the tourism service functions with their own characteristics (such as a long history and cultural characteristics, slow life rhythm city, seafood cities, etc.). At the same time, these cities should strengthen contact with their edge cities, build a tourism service platform for them, and enhance the influence of regional tourism services with them.
Finally, tourist cities with low-hierarchical tendencies (edge cities) need to improve their urban tourism service functions, form a more comprehensive service reception capacity, and expand the number of tourists received, so as to increase tourist satisfaction. At the same time, these cities need to actively pay attention to the influence of the administrative power of Chinese cities and create their own tourism service characteristics with the help of tourism planning (such as healthcare locations, holiday resorts, natural landscape wonders, etc.), to gradually establish their own tourism image in the regional tourism scope.
In the "discussion and conclusion" part of the original manuscript is simplified to highlight the contents of conclusion, so this part is mainly adjusted to the conclusion. In the revised version, I clearly highlighte the novelty of the study in the last paragraph of the introduction:
We find that the interlocking network model of tourism services also shows the geographical structure of "center-edge" in reality, but it is a new "center-edge" mode. Its novelty lies in: first, the new "center-edge" model is a polycentric three-dimensional connectivity network, central cities and edge cities have different hierarchical tendencies, and they also have different functional roles in the network.second, the new "center-edge" model is bidirectional flow, unlike cities in the traditional model, edge central and edge cities can provide tourism service products and tourists with each other. Therefore, the new model in this paper would enrich the application of "center-edge" theory in new research scenarios.
Not only that, I also elaborated on this novelty in the second part of the discussion:
Therefore, it can be seen that there will still be a "center-edge" regional distribution structure in China's overall tourism service network. With the maturity of the tourism service network, the position of the central cities may be strengthened, while the edge cities may be reduced. The new "center-edge" model based on tourism service network is a theoretical improvement for the "center-edge" theory in traditional geography. The novelties are shown in: first, the new "center-edge" model is a polycentric three-dimensional connectivity network,the network can be divided into four different hierarchical tendencies, and they also have four functional roles in the network. In the Fuzzy c-means cluster analysis, we find that there are hybrid tourist cities, and we found that different "center-edge" clusters or arenas also have the possibility of cross group interaction, which are closer to the reality of tourism services. Second, the new "center-edge" model is bidirectional flow, and the central city (high- hierarchy cities) can provide tourism service products and tourists to edge cities (low- hierarchy cities), and edge cities can also provide tourism service products and tourists to high- hierarchy cities. However, the traditional "center periphery theory" often thinks that the edge cities provide agricultural products or development capital to the central cities, while the central cities provide industrial products or services for the edge cities. This flow process can’t flow in two directions at the same time.
Point 3: Furthermore, in Line 26: Castells’ theory: Please provide here a brief description of this theory or at least a relevant reference.
Response:Thank you very much for your suggestions.
In the beginning, I directly quoted Castell's theory, and pointed out the shortcomings of his theory in tourism service:
One of most important academic viewpoints in Castells’ theory pointed out: “the deep combination between the information technology revolution and capitalist reorganization constructs the flow network social space. The city is a spatial unit of labor reproduction or a space fragment in the network society, and the network society is composed of multiple cities[1]”.
He also think that network society relates to such practice phenomenon that important tourist cities across some specific large regions are used by different capitals as “basing points” in the connection network of service and production [2].
The resulting connections make it necessary to promote the development of important cities and arrange some service-oriented tourist cities into a connected network hierarchy. However, the lack of theoretical agreement on the defining characteristics of tourist cities in Castells’ theory that perform their important tourism service functions has resulted in scientific taxonomies, may usually limited to focus on the highest hierarchical cities.
In the original manuscript, the first and second references quoted the results of Castells. Because of the reference format required by the journal of Sustainability, castells is abbreviated as C, so it is not easy to find, in order to facilitate readers' understanding, I changed the "castells" at the beginning of the introduction to "Manuel·Castells" in the new revised version.
- Manuel, C. The Internet Galaxy: Reflections on the Internet, Business, and Society. Quarterly Review of Distance Education 2004, 5, 66-68.
- Manuel, C. Rise of the Network Society. Blackwell Publishers: New Jersey, 2000; p.389-414.
Point 4: Lines 52-54: this paragraph is not clear. Please improve English form.
Response: Thanks. I've adjusted the expression (Lines 52-54).
For example, Beijing as an international service-oriented tourist city, as the capital of China, it is the urban function centers of China's political management, cultural production, high educational supply, and passenger transport hub. these important urban functions prompt many tourists will most likely prefer to travel to Beijing. Compared with Beijing, Although Guilin has its high-quality natural landscape tourism resources, but it has not many types of urban functions, and the city’s financial function, exhibition function, cultural production, passenger transportation and other service functions are not powerful. Under the restriction of the city's service functions, it can’t attract many tourists with high-end tourism consumption needs to flow into the city, driving the city to generate considerable tourism economic benefits. so it does not occupy an important hierarchical in the tourism connection network.
Point 5: In the literature review section: a great part of this section, should be included in the Methodology section because it explains the theoretical framework adopted in the study. In the literature section the authors should include the relevant literature that regards research study similar to that analysed in their research, highlighting any points of weakness that will be overcame by their research work. This is crucial to improve in the introduction section where the novelty of the research should be better described (see lines 65-67).
Response: Thank you for your suggestion, I have made corresponding changes in the literature section:
First: in the revised paper, section 2.2 has been added some literatures on discuss the methodological deficiencies in the previous cluster classification;
The details are as follows:
Cluster analysis is one of the most popularly empirical analysis techniques to study the data matrix of a very large number of tourist cities[1]. In this paper, the traditional clustering algorithm is applied to the relation matrix generated based on the connection between various cities, and then the mutually exclusive city clustering is calculated according to the “tourism service value” of the service-oriented tourist city in the network. But this classical clustering analysis method is full of various problems in practical application [10] :
Firstly, Camille Roth et.al. pointed out that the connected network of various tourist cities does not show a simple and absolute hierarchical city structure[11]. In fact, the connections between different tourist cities are a complex network rather than a simple hierarchical structure [11]. This is an empirical support for the opinion in Taylor's paper, he believes that the clear patterns provided by traditional clustering algorithms are unlikely to provide an accurate and sensitive specification in a complex city network[13].
Second, Skelcher Chris believes that functional patterns and hybrid hierarchy increase the complexity of network[14]. When the networks of various scales are gathered together in a large region, the result is a complex network structure, showing multiple regional patterns and hierarchies. Therefore, the exploration of service-oriented tourist city network should include the evaluation of the hierarchical structure of overall region and the intertwining of regional patterns. Similarly, traditional crisp cluster classification analysis is unlikely to provide sensitive specifications for the scrambling patterns [15].
Third, Taylor Peter found that the previous classification evaluation is limited to the medium and upper rungs of the urban hierarchy system[15]. The main reason is that the outer areas of the urban network will be classified based on sparse data, so any classification based on the data will yielding vagueness[17]. Therefore, minor changes in sparse data usually yield completely different classification results, while mutually exclusive cluster classifications will not be unbiased in theory.
Second, Methodology 3.4 added the description and introduction of the Fuzzy C-means method.
Third, Based on your suggestions, I summarize the novelty of the study in the Introduction section:
Because of the connected network closely related to the geographical distribution of cities, the novelty of the study must attempt to describe and explain the connected model of interlocking network based on tourism services from the geographical perspective, and point out the geography “gap” faced by service-oriented tourist city network. In terms of Methodology, we found that there are some problems in the previous cluster analysis, for example, due to the dynamic development of tourism, some cities do not belong to a certain cluster. Therefore, one of the specific novelties is to use Fuzzy c-means clustering analysis to solve these problems. We find that the interlocking network model of tourism services also shows the geographical structure of "center-edge" in reality, but it is a new "center-edge" mode. The other specific novelties lie in the following aspects: first, the new "center-edge" model is a polycentric three-dimensional connectivity network, central cities and edge cities have different hierarchical tendencies, and they also have different functional roles in the network. second, the new "center-edge" model is bidirectional flow, unlike cities in the traditional model, edge central and edge cities can provide tourism service products and tourists with each other. Therefore, the new model in this paper would enrich the application of "center-edge" theory in new research scenarios.
Point 6: In the section methodology: 3 please move here part of the text reported in section 2, i.e., all the part which are used ad method for your study. Furthermore, since the methodology framework takes into account several steps described in the previous section, in order to improve the readability of the paper, the Authors should provide a flowchart to describe the different phase of the method. In the present form section 3 seems to describe a mere case study and it does not look like an original method to overcame limitations that should be well described in the introduction section.
Response: Thank you very much. This question did cost me a lot of energy and time. After reading a lot of references, I basically adjusted this part of the manuscript.
First, I’ve remove “tourism connection network” in the line 149 of original paper, and I adjusted the sentence statement of this position.
The rise of the tourism connection network is one of the most fundamental changes in the current tourism development.
Second,the literature part 2.2 (Fuzzy c-means cluster analysis ) of original submitted paper is adjusted to the part 3.4 of revised paper, which is to improve the explanation of the Methodology section.
Third,the literature part 2.7 of original submitted paper is adjusted to the part 3.5 of revised paper, which is to improve the explanation of the Methodology section.
Fourth, your suggestion is very correct, thank you. I sort out the literatures of cluster analysis method, mainly to find the problems reflected in the previous research on the relationship matrix data of city connectivity, and to answer why fuzzy c-means clustering analysis method is suitable for the solution of the research problem in this paper. At the same time, I sincerely tell you that I have been staying up late and busy with the task of revising articles for more than ten days because I am worried that your suggestion will not be met. Reading the method references to describe the different phase of the method is indeed a big challenge for a non-native foreign language author, but I think from your suggestions, the quality of my articles has been improved. I believe I will work hard in the future, and this needs to be obtained your support and understanding.
By the way, inspired by your request to make a flowchart, in order to explain the problem in detail, I also add two pictures in revised article, one is a taxonomy of clustering approaches; the other is fuzzy C-means cluster analysis based on the needs of the research method.
Figure 1 A taxonomy of clustering approaches (Jain, Anil K,1999)[32]
Fig 2 Fuzzy c-means cluster analysis (Jain, Anil K,1999)[32]
Point 7: Section 3.2 data production: line 275: Credible? Please clarify! What do you mean with credible?
Response: Thanks. I very much agree with your suggestion.
This method, I mainly learned from Derudder and Taylor's research method in GaWc
(Derudder, Taylor, Witlox,Catalano, Hierarchical Tendencies and Regional Patterns in the World City Network: A Global Urban Analysis of 234 Cities. Regional Studies, 2003, 37(9):875-886). In their data collection content of Methodology section. They believe that:
“Information for every firm was simplified into ‘ values’ ranging from 0 to 5 as follows. The city housing a firm’s headquarters was scored 5, a city with no office of that firm was scored 0. An ‘ordinary ‘ or ‘typical’ office of the firm resulted in a city scoring 2. With something missing (e.g. no partners in a law office), the score reduced to 1. Particularly large offices were scored 3 and those with important extra-territorial functions (e.g. regional offices) scored 4. The end-result was a 316×100 matrix Vi j, where vij ranges from 0 to 5.”
After using the data production method, I found that the research results (the data in the attachment of the original manuscript provided by me, file name: “the standardized value of network connectivity in the 63 cities”) conform to the general judgment of our expert team on the hierarchical tendencies of China's service-oriented tourist cities, without great deviation and ambiguity.
Point 8: lines 279-284. "0 means there is no distribution, 2 as the normal conditions, and 5 as the highest " Please, justify this sentence is not clear. Are any relevant studies that support this in literature?

Response: Thank you for your comments. I justify the sentence to make the expression clear:
“0 means that the city does not have a specific organization (one of the 38 typical tourism organizations), 2 represents the city has the subsidiary company of a specific organization, and 5 represents the head company of a specific organization in the city.”
There are a relevant study of literature that support this data production method,which is used by Derudder and Taylor's research method in GaWc (Derudder, Taylor, Witlox,Catalano, Hierarchical Tendencies and Regional Patterns in the World City Network: A Global Urban Analysis of 234 Cities. Regional Studies, 2003, 37(9):875-886). In their data collection content of Methodology section.
Point 9: Section 3.3 data collection: line 316: “Precise specifications guide our data collection process relatively smooth “. Please, clarify.
Response: Thank you for your question.
This expression is based on a more optimistic description. In fact, I spent six days on the websites of 38 advanced tourism service organizations' head offices and their subsidiaries to obtain data (see my attachment: "WCitiesSub" in the folder "Non-published Material" ). Objectively speaking, this process is a relatively arduous process. Academic research requires such efforts, but in the end we successfully completed the data collection goal, and the original data did not have a large area of missing.
In order to avoid ambiguity, I adjusted the sentence:
Precise specifications guided the data collection process here, which I finally complete the task of data collection after hard work.
Point 10: Section 5. Discussion and Conclusion: This section is more conclusion than discussion. It misses discussion with other research papers in order to highlight the novelty of this study. The authors cite only the paper that developed the method adopted. In the present form, this research is a case study.
Response: Thank you for your comments.
Your suggestion is quite correct, so I will separate the discussion from the “5. Discussion and Conclusion” in the original submission, and write the discussion and conclusion separately:
I rewrote all of the discussion, In the Disscusion:
the first part mainly discusses the author's judgment on the future development tendencies of tourism network model based on empirical discussion;
the second part discusses the novelty of the "center edge" structure of tourism service connection network;
the third part discusses the essential reasons for the difference of tourism service connectivity, and the reasonable development measures of different levels of tourism cities.
At the same time, I simplified the conclusion to make the expression more accurate:
Based on Manuel·Castells' theory of the flow space network [2], this paper has enriched the theory of city position and classification in a flow space network using fuzzy c-means clustering analysis. In addition, from GaWC's research [28], this paper has developed an empirical method to study the formation of tourist city networks. The method improves the spatial structure theory of tourism geography in a relatively complete large national region and especially complements the “center-edge” theory.
The results here obviously reveal the differences between connectivity and geographical distribution from the data analysis, indicating the regional pattern characteristics and hierarchical tendencies distinctions of cities in the network. These hierarchical tendencies and regional patterns prompt three conclusions:
Firstly, China’s four most important international cities, as central structures, lead the whole tourism connection network, and the connectivity of each city is very high. As the first hierarchical group of cities, these four cities have formed a unique tourism arena. They all have their own unique urban functions, and their tourism services are not easily replaced by other cities. In terms of their geographical distribution, they occupy the northern, eastern, and southern regions of China. This spatial layout can effectively connect them to more low-hierarchy cities in the geographical space. At the same time, there is an obvious phenomenon in the interaction between hierarchical tendencies and regional patterns, that is, clusters with low average connectivity are more restricted by regional forces in special regions, and they have less influence on cities that are farther away. This geographical phenomenon further shows that a regional pattern is not only the location space of different clusters, but also represents the interaction results of various tourism arenas in the urban network with different hierarchical tendencies.
Secondly, the regional patterns show that eastern, southern, and northern China are the three most important arenas of the tourism service economy in China, but there are great differences in the configurations of these arena. In northern China, the configuration is different from eastern China and southern China in the following aspects: (1) There is a “vertical” structure of service connectivity in tourist cities, and there is no second hierarchy service-oriented tourist city. Beijing's comparative advantages and scale of service connectivity are both far stronger than the surrounding tourist cities. (2) In the process of tourism service globalization, high-quality urban functions are excessively concentrated in the nuclear cities. In contrast, the tourist cities in eastern and southern China have better service connectivity on the whole, and their cities cover all arena hierarchies. The connectivity scale distribution structure of the tourist cities of the two regions tends to be “horizontal”, and tourist cities in eastern China perform better. The two regions of southwestern and central China are similar in terms of their tourism service network. Chongqing and Chengdu in southwestern China are more developed than Changsha and Wuhan in central China, but the overall development level of central China is more ideal. At the same time, this paper highlights the former “previously easily neglected tourist cities” in the service-oriented tourist city network, and northwestern and northeastern China are well-represented in the network beyond the main service regions. Xi'an and Hohhot are relatively important representatives of the tourism services in northwestern China, and Dalian and Shenyang have gathered a strong performance within northeastern China.
Thirdly, this paper has attempted to improve the understanding of tourism regionalization by using a new and detailed description of a tourism connection network based on regional patterns. Instead of limiting “region forces” to several major “international cities”, this work incorporated a very large number of various tourist cities into a single integrated region to construct an analysis framework for urban tourism connections. The geographical distribution of contemporary tourism service globalization in the regional scope is not an end-product, but an important part in a series of continuous processes. This means that as the connectivity of the service-oriented tourist city network intensifies, the gaps identified in the network will be filled up in the next few years. On the other hand, with the development of a regional growth pole at the national level, the regional service industry of urban tourism may be more and more concentrated in a few important tourist cities, and the gaps of the tourism connection network may widen. Due to the rapid development of China's urban tourism, these hierarchical tendencies and regional patterns of the future tourism network should also change. In the face of such unknown changes, we should have a scientific understanding of the contemporary urban tourism network, otherwise we will not be able to evaluate these changes, nor can timely and correct updates be made to the strategies of urban tourism development in China.
At last, highlight the novelty of this study in the “introduction section”, and explain it in detail in the “discussion section”.
Therefore, one of the specific novelties is to use fuzzy c-means clustering analysis to solve these problems. We find that the interlocking network model of tourism services also shows the geographical structure of "center-edge" in reality, but it is a new "center-edge" mode. The other specific novelties lie in the following aspects: first, the new "center-edge" model is a polycentric three-dimensional connectivity network, central cities and edge cities have different hierarchical tendencies, and they also have different functional roles in the network. second, the new "center-edge" model is bidirectional flow, unlike cities in the traditional model, edge central and edge cities can provide tourism service products and tourists with each other.(Note:where it is in the last paragraph of the introduction section in the revised version of the paper.)
Thank you for your understanding and support for the paper. When the article is about to be revised, I would like to express my my best wishes to you for these valuable comments and suggestions!

Round 2
Reviewer 1 Report
Tkank you for detailed comments
Author Response
Dear reviewer1:
Thank you for your understanding and support. I am surprised to find that you don't give me some specific comments and suggestions in the “Comments and Suggestions for Authors
” this round, which makes me very grateful. This is an encouragement for me.
However, I know that my paper is still not so perfect. I also see that you have made an overall evaluation of the paper, there are two items are “can be improved”.
so, there are two brief explanations in the attachment. Please see the attachment.
Kind regards.
Sincerely yours,
Author

Reviewer 2 Report
Dear Authors thank you for the revision of the manuscript. The paper improved however, some other changes are required before possible publication.
-
- In general, throughout the manuscript please avoid first person. “I”, “We”..and so on is not a polite way to describe a research study. For example in line 329 “ Precise specifications guided the data collection process here, which I finally complete the task of data collection after hard work”. Moreover, is not relevant to highlight the hard work, since generally the work is the research activities.
- The English form has again some critical troubles. For example, please read Line 86-9. The sentences have great problem in the structure of the text. Please revise English of the text between these lines and of the whole manuscript.
- Again, Line 283 “Credible”? Please clarify! What do you mean with credible? Maybe the word “Credible” could be changed in “More adherent with general judgment of expert team on the hierarchical tendencies of China's service-oriented tourist cities”.
Author Response
Dear reviewer2:
Thank you for your valuable comments on my paper. I can see that you have read it carefully. In this round of revision, I made corresponding modifications and adjustments in response to your comments and Suggestions.
Please see the attachment.
Kind regards.
Sincerely yours,
Author
